# Application of Non-Contact Infrared Monitoring Technology in Marine Controlled-source Electromagnetic Transmission System

Chentao Wang[1], Ming Deng[1], Zhibin Ren[1], Meng Wang[1]

[1]China University of Geosciences, Beijing, No. 29 Xueyuan Road, Haidian District, 100083

*Correspondence to: Meng Wang (wangmengcugb@qq.com)*

**Abstract.** In recent years, marine controlled-source electromagnetic (MCSEM) systems have become increasingly crucial for offshore resource exploration. However, the high-power operation of these systems poses significant safety challenges, mainly stemming from undetectable thermal anomalies. This study introduces a novel integration of non-contact infrared thermal imaging technology into MCSEM transmission systems, superseding conventional point-contact temperature measurement methods with comprehensive, real-time surface thermal monitoring. The proposed system effectively resolves several critical issues specific to MCSEM operations, particularly electromagnetic interference (EMI) resilience, high-temperature operational stability, and data transmission bandwidth limitations. Our hardware-software co-design methodology achieves dual optimization of measurement efficiency and operational safety. Hardware advancements incorporate Gigabit Ethernet for enhanced data throughput, EMI-resistant circuitry for improved signal integrity, and motorized zoom lenses for adaptive infrared imaging capabilities. Concurrently, our software architecture facilitates real-time thermal visualization, robust offline data storage, and intelligent region-of-interest temperature alert mechanisms. This research establishes a new operational paradigm for MCSEM monitoring systems, significantly enhancing safety protocols and enabling proactive risk management in high-power offshore applications.

**Keywords:** MCESM; Infrared Thermal Imaging; Hardware-Software Co-Design; Real-Time Monitoring; Offshore Safety

## 1 Introduction

The marine electromagnetic method, which initially emerged as a specialized field in the 1970s primarily for military detection purposes, has evolved significantly over the decades. This evolution was driven by the recognition of its economic potential, particularly in marine oil and gas exploration applications (Constable, 2010). Marine electromagnetic technology has matured into a crucial resource exploration tool (Constable et al., 1986; Kasaya et al., 2023; Menezes et al., 2023). As shown in Figure 1, a standard MCSEM system consists of two primary components: (1) electromagnetic transmitters and (2) seabed receivers. In this method, electrical anomaly bodies react to the electromagnetic waves emitted by the transmitter, and the receiver captures and analyzes these responses to pinpoint the location and attributes of the anomaly (Constable and Srnka, 2007).

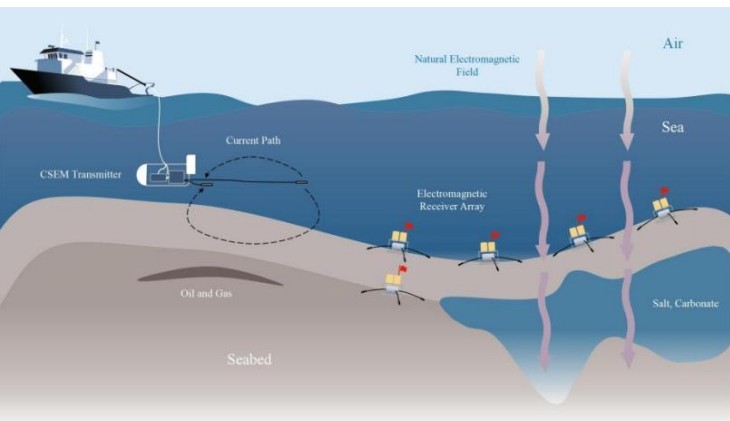

**Figure 1: MCSEM instruments and offshore operations.**

The current developmental emphasis of the MCSEM transmission system lies in high-power applications, which introduces a spectrum of security challenges. Addressing these challenges effectively necessitates a thorough comprehension of the prevalent high-power components integral to the transmission system. In the transmission system conceptualized by Wang et al., (2022), the high-power constituents predominantly encompass the power located at the deck end and the transmitter situated at the underwater. In the development of MCSEM, researchers have established several monitoring methods to ensure the safety of its

offshore operations. Deng et al., (2013) paved the way for data acquisition in deep-water magnetotelluric studies, emphasizing the importance of collecting voltage and current data within the transmitter. Simultaneously, Wang et al., (2013) developed the MCSEM deck monitoring system in the same year, employing a serial port for data transmission, and the deck monitoring system was utilized in marine experiments in 2013, tracking indicators such as emission current, bus voltage, cabin temperature, and battery voltage. Xu et al., (2017) introduced a remote monitoring device for MCSEM, utilizing photoelectric composite

cable technology. This device essentially facilitated the transmission of serial port information through a conversion module. Deng (2022) applied the electric insulation online monitoring unit into the marine experiments. Following this, Wang et al., (2023) developed an online transmission system for the complete current data file of the transmitter, significantly enhancing data transmission efficiency. In the newly designed 2000A-class high-power transmitter by Wang et al., (2024), the critical role of temperature control in marine controlled-source electromagnetic (MCSEM) transmission systems is further highlighted.

Nevertheless, despite these advancements, the primary monitored parameters in these systems remain limited to voltage and current, while temperature monitoring is only implemented at selected nodes through contact-based measurements. During a marine experiment conducted in the South China Sea in 2023, observable black marks indicative of high temperatures were found in the deck power booster unit and transmitter, as depicted in Fig. 2a and Fig. 2b. Unfortunately, the existing monitoring systems failed to detect these anomalies, which not only posed significant safety hazards but also ultimately led to substantial

economic losses.

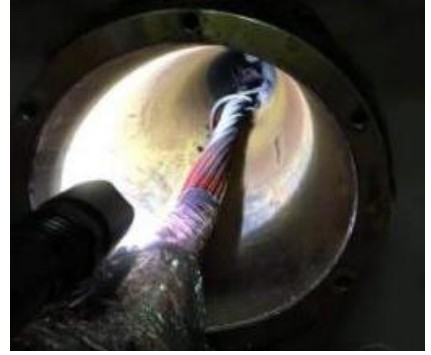 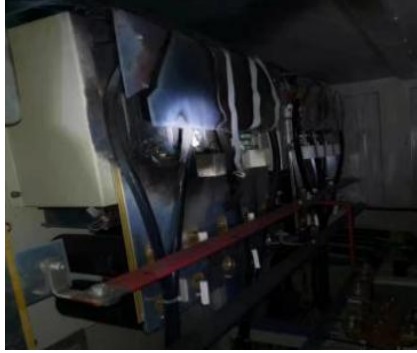

**(a) Burnt shipboard transmission cable**     **(b) Power booster overheating**
**Figure 2: The risks of thermal overload in MCSEM system.**

    In summary, the existing MCSEM temperature monitoring system exhibits several limitations. Conventional temperature measurements have primarily employed contact-based sensors such as PT200, typically deployed at a limited number of internal

nodes within the transmitter. However, during high-power operations, critical risk areas extend beyond these nodes to encompass the power supply unit, the photoelectric composite cable, and various internal regions of the transmitter. Non-contact infrared measurement offers benefits such as a wide recognition area, fast response time, safety in use, and long lifespan compared to traditional contact-based systems (Shen et al., 2018; Sullivan et al., 2021). Despite these advantages, currently available infrared surveillance equipment lacks compatibility with MCSEM systems, particularly in addressing the unique challenges presented by

harsh MCSEM operating conditions. The MCSEM operations are often subjected to intense electromagnetic interference and

extreme temperature fluctuations, which can severely compromise the reliability and accuracy of conventional monitoring devices. Moreover, seabed-deployed equipment must incorporate substantial offline storage capacity to maintain continuous data recording during unexpected disconnections, which frequently occur in deep-sea operations. Additionally, both data transmission speed and methodology require meticulous design to ensure seamless integration with existing marine electromagnetic infrastructure, ensuring uninterrupted and efficient data flow.

To address these challenges, we developed a novel non-contact temperature monitoring system for MCSEM applications through comprehensive optimization of electrical circuits, mechanical configurations, and optical architectures. Corresponding temperature early-warning algorithms and monitoring software were also developed. This innovation transforms the traditional contact-based point measurement approach of marine controlled-source electromagnetic (MCSEM) systems into a non-contact surface measurement paradigm. The proposed system significantly expands the monitoring coverage, enhances measurement efficiency, and ensures both data integrity and operational safety in marine environments.

## 2 Instrumentation design and multi-node deployment

As shown in Figure 3, multiple critical power nodes are distributed throughout the MCSEM offshore operation system, spanning the deck, ship's hold, and underwater transmitter. However, the traditional contact-based temperature measurement method is only applied to the IGBT nodes inside the underwater transmitter. This single method is clearly insufficient to meet the safety requirements of offshore operations. Therefore, it is essential to deploy suitable temperature measurement systems at each of these key nodes.

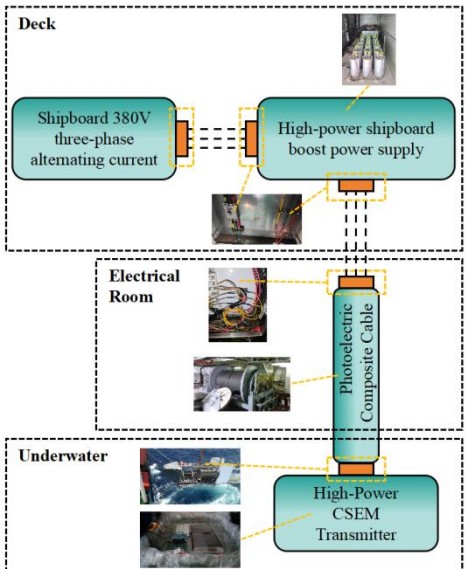

**Figure 3: Key power nodes in MCSEM**

## 2.1 Hardware Design Scheme

The hardware circuit design is centered around the PLUG617 infrared core provided by GSTiR company. The parameter mode of the scene source data supported by PLUG617 is Y16 + parameter line, with a resolution of 640*513 and a frame rate of 30Hz, the required data transmission rate is at least:

$$640 \times 513 \times 16bit \times 30 / s = 150.3Mbps \tag{1}$$

Consequently, Gigabit Ethernet constitutes the minimum requirement to satisfy both the video data transmission rate demands and control signal communication needs. Furthermore, MCSEM operational constraints necessitate local data storage within

underwater equipment to ensure data security. In traditional marine electromagnetic instruments, conventional microcontrollers, such as STM32F4 Series, are typically employed. However, such microcontrollers lack the computational throughput and memory bandwidth required for: (1) high-volume video data transmission, and (2) large-capacity storage access - both critical for video transmission system functionality. To address these limitations, we have adopted the A40I platform from Forlinx and installed the Linux operating system to fulfill our needs. The hardware circuit design primarily focuses on transmission speed, storage capacity (Sun et al., 2022). The system architecture is illustrated in Figure 4.

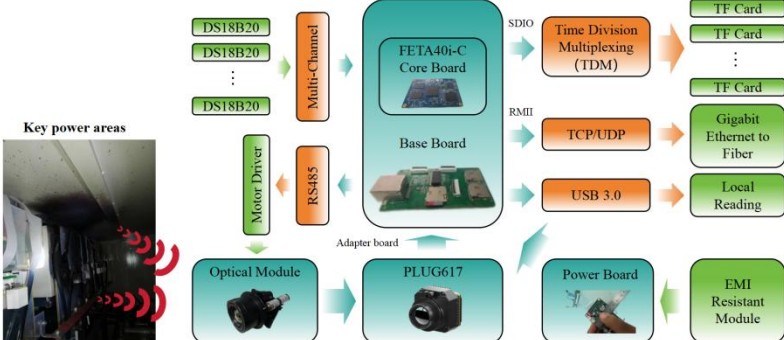

**Figure 4: Hardware circuit structure**

The primary hardware architecture of the system encompasses three circuits: the main control board, the power supply board, and the A40I platform. To facilitate the zoom functionality of the infrared lens, an additional motor system has been integrated to control the optical module, which is connected to the main control board via a 485 interface. Moreover, leveraging time-division multiplexing technology, the system is capable of accommodating multiple high-capacity TF cards for offline storage. Given that the internal circuitry of the instrument generates a significant amount of heat during extensive data production, which is non-negligible, multiple DS18B20 sensors have been deployed within the device for temperature monitoring and subsequent calibration of temperature data. The system communicates with external devices through an RMII interface for gigabit Ethernet connectivity, and data is transmitted over long distances via a network-to-fiber module.

## 2.2 Anti-Electromagnetic Interference(EMI) Design

During operation, marine electromagnetic transmitters generate substantial currents ($10^2$-$10^4$A), creating a potent source of electromagnetic interference (EMI). Ensuring the integrity and accuracy of large amounts of data transmission in such close proximity to such a strong interference source is a formidable challenge. Duan et al., (2018) utilized CAN isolation as a measure to tackle EMI affecting communication lines, thus partially enhancing communication stability. Nevertheless, it is crucial to recognize that EMI incidents commonly manifest at various levels, necessitating the management of both external interferences on internal circuits and self-generated electromagnetic disturbances.

The sources of electromagnetic interference are illustrated in Figure 5. Although the system's metallic enclosure and metallic shielding mesh cover can largely shield electromagnetic interference, interference may still couple onto the connection lines between the circuits and the external environment. Therefore, we have implemented anti-electromagnetic interference designs for both the power input interface and the Ethernet interface. Figures 6 and 7 showcase the design of our system's power input and data output interfaces. The design in Figure 6 primarily includes: (1) Placing a GDT (Gas Discharge Tube) at the first stage to dissipate instantaneous high voltage, with capacitors used to filter out common-mode and differential-mode interference. (2) Positioning a common-mode inductor at the second stage and constructing a feedback loop with a GDT to further filter out common-mode interference introduced from the outside. (3) Placing a TVS (Transient Voltage Suppressor) at the third stage to

eliminate residual energy not absorbed by the previous stages, ensuring the output voltage remains within a reasonable range.

(4) Common-mode filtering capacitors and differential-mode filtering capacitors are employed to suppress different types of noise interference. In Figure 7, we employ network transformers to divide the circuit into digital and analog ends, isolating interference while enhancing the signal. Additionally, a GDT is introduced at the later stage to prevent instantaneous high currents. This approach enables longer transmission distances while isolating the chip end from the external environment, thereby enhancing anti-interference capability.

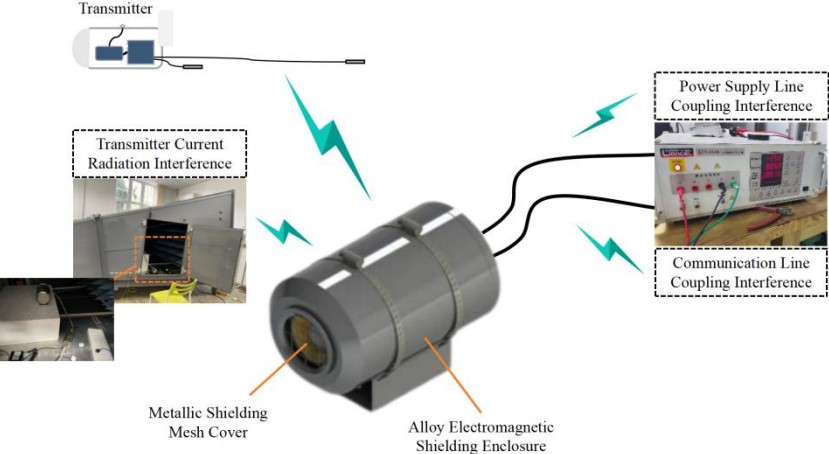

**Figure 5: Electromagnetic interference sources**

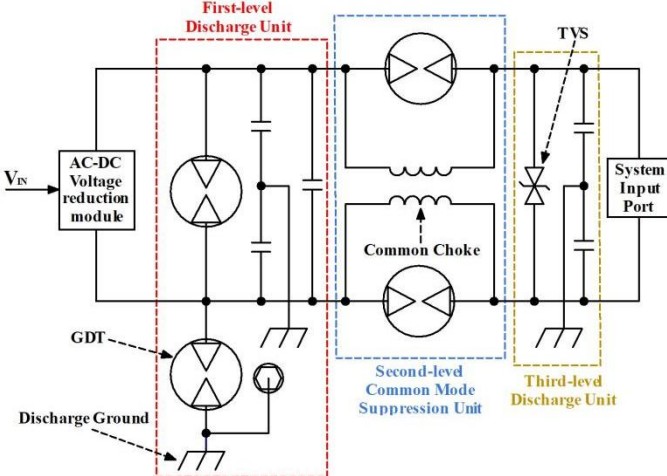

**Figure 6: The power interface design.**

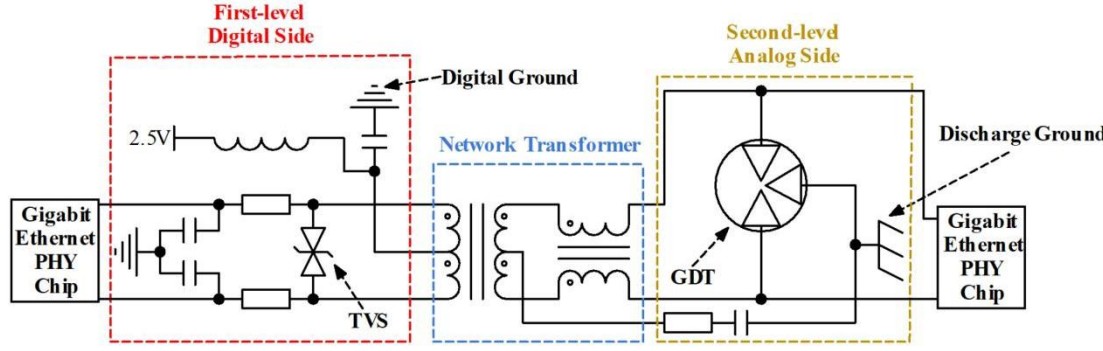

**Figure 7: The data interface design.**

## 2.3 Optical System Structure Design

In addition to the accuracy of the infrared sensor itself, the accuracy of the optical system will also introduce certain errors. Therefore, we carry out precise measurements and optical recalculation after the processing of optical components and structural components to reduce the impact of processing errors. At the same time, according to the sensitivity analysis results, a reflective collimator is used for the assembly of components with relatively strict requirements to ensure assembly accuracy. The overall structure of the optical system is shown in Figure 8, which adopts a 4-lens and 5-aperture structure. The external support structure is made of 2A12 aluminum alloy to reduce its weight, and it can withstand a maximum stress of 102MPa (The occurrence site is at the interface of the electrical box installation.). Ensure that the component tilt is less than 2 arcminutes, the eccentricity of the original component is less than 0.03, and the surface tilt is less than 0.015 degrees. Modulation transfer function (MTF) and spot diagrams were used for aberration performance analysis of the optimized system (Zhang et al., 2022). Pixel pitch of the infrared sensor we use is 17, so the Nyquist frequency N is:

$$N = 1/2a = 1 Linepairs/(2 \times 17\mu m) \approx 29.14 lp/mm \tag{2}$$

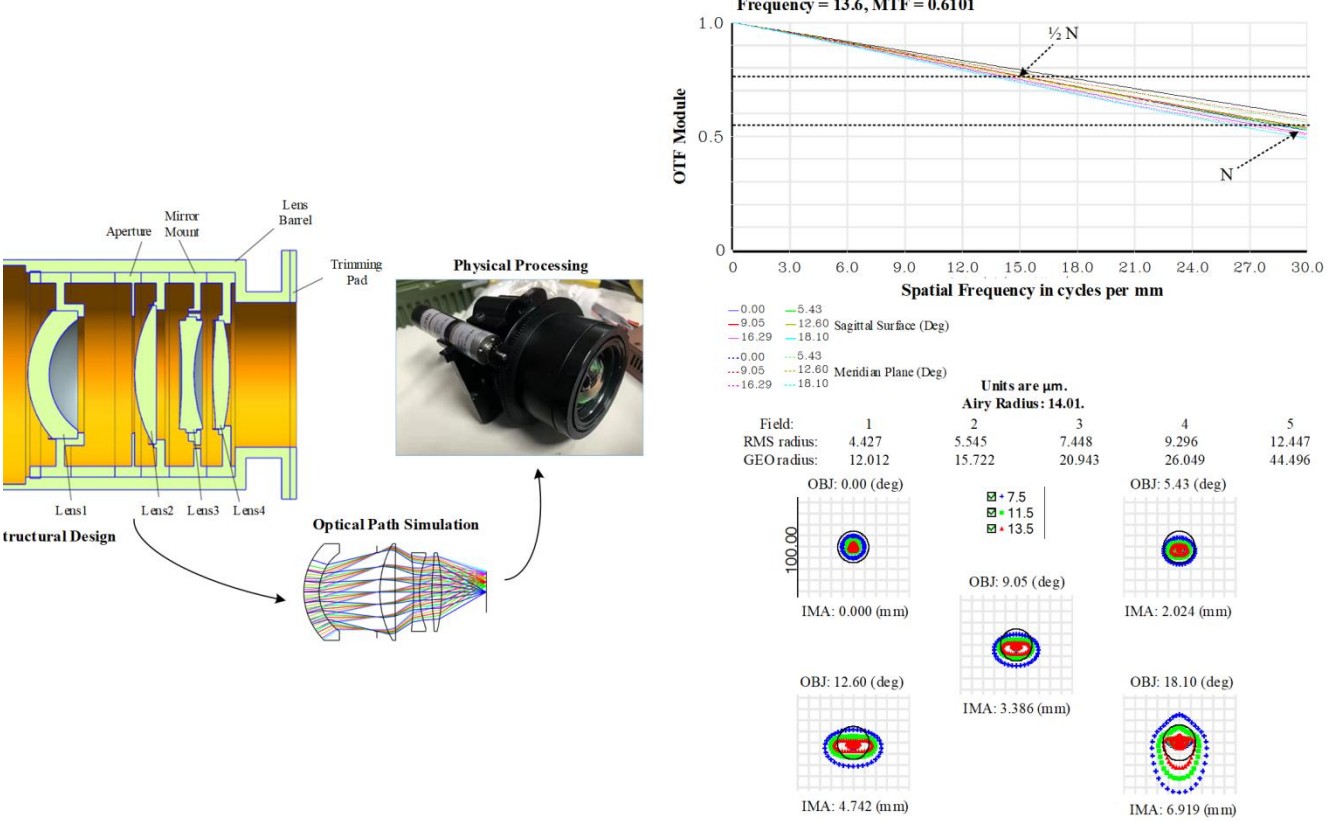

Figure 8: The data interface design.

Where $a$ is the pixel size. As shown in Figure 11a, at the 1/2 Nyquist frequency of 15 lp/mm, the MTF value is greater than 0.7, and MTF value in the full FOV (Field of View) is greater than 0.5, and the lens can achieve fairly good performance up to 30 lp/mm. The Airy spot radius is 14.01 μm, the RMS spot radius of the system is 4.427 μm, which is within acceptable error. The imaging quality reaches close to diffraction limited performance, and the imaging performance is uniform across all FOVs and the optical system has good energy collection in various FOVs.

## 2.4 Mechanical Structure Design and Stability Verification

Previous studies have demonstrated that measurement distance significantly affects the accuracy of infrared temperature measurement systems (Long, 2016; Machado et al., 2024). To mitigate errors stemming from distance, we plan to incorporate a zoom system into our design. By incorporating a zoom lens controlled by a motor into the infrared sensor, as illustrated in Figure 9, we aim to enhance the precision of our measurements. Furthermore, the integration of a zoom system brings increased flexibility in deploying our equipment. The motor communicates with the main control board through RS485. This allows real-time monitoring of the motor's status and position. Due to high-speed data transmission and processing, chips generate a significant amount of heat. If this heat is not properly managed, it can lead to device malfunction. Therefore, we utilize copper plates and thermal grease to conduct the heat from the chips, as depicted in Figure 9.

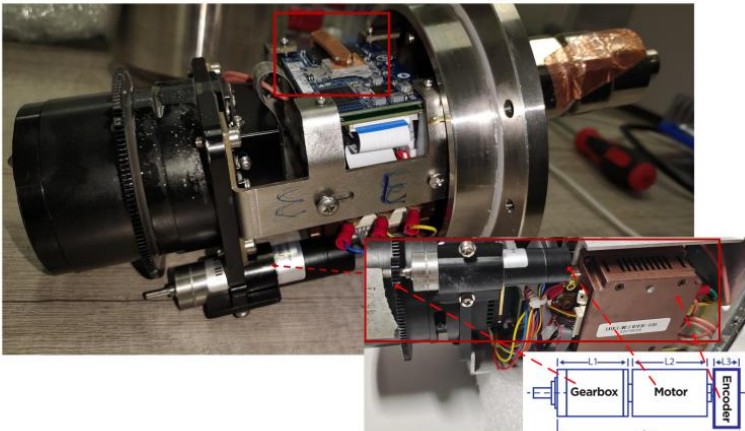

**Figure 9: Optical Zoom and Heat Dissipation Design.**

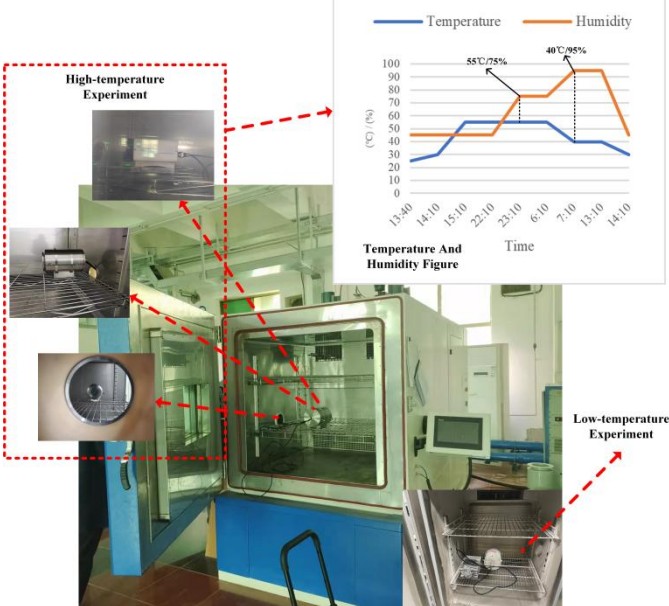

**Figure 10: High-temperature and high-humidity testing**

In order to evaluate the system's resilience in hostile environmental conditions, we began by subjecting the equipment to temperature and humidity tests, as depicted in Figure 10. The instrument underwent long-term performance assessments at both 55°C and 75% humidity, as well as at 40°C and 95% humidity, confirming its operational stability. Moreover, we explored the

system's adaptability to low temperature and high humidity settings. Notably, the image captured frost formation on the casing of the instrument.

In conclusion, we have successfully adapted the infrared system for MCSEM applications through comprehensive modifications in circuit design, optical system configuration, and mechanical housing, accompanied by rigorous testing. This adaptation ensures reliable operation of the infrared system under the challenging conditions of MCSEM deployment, including harsh electromagnetic interference, elevated temperature and humidity levels, and diverse environmental settings. Nevertheless, to fully realize multi-node synchronization, efficient data processing, and functional implementation via the existing MCSEM communication infrastructure, additional software development remains necessary to complete and optimize these operational aspects.

## 3 Software development and data processing

The software is an important component of the entire infrared temperature measurement system, including embedded software and user-oriented operation software on the PC side. Hermann et al., (2020) released an open-source hardware and software for an infrared system, primarily designed for offline storage-based infrared monitoring of underwater objects. Yu et al., (2023) also provided a real-time temperature measurement solution, which achieved real-time identification of characteristic phenomena changes through hardware data collection and online software analysis. Beyond conventional infrared monitoring capabilities, our offshore deployment scenario necessitates robust features including: (1) offline data caching with scheduled upload, (2) autonomous connection recovery, and (3) region-of-interest alert mechanisms. These advanced operational requirements mandate tight coordination between the embedded control software and PC-based management interface.

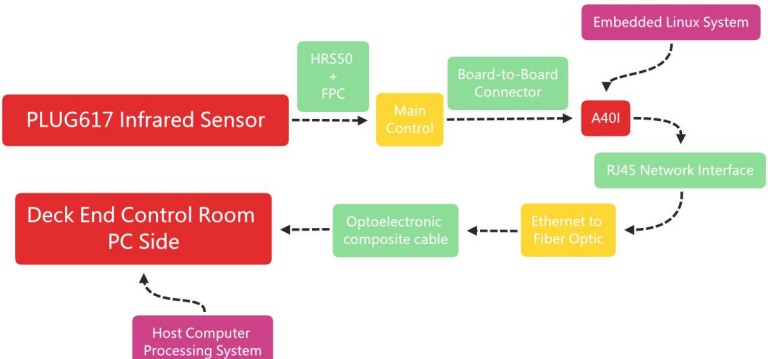

**Figure 11: The data flow of the infrared system.**

Figure 11 illustrates the data flow of the infrared system. Initially, data generated by the infrared sensor is transmitted to the main control board via the HRS and FPC connectors. The main control board subsequently transfers this data to the A40I processor via board-to-board connectors. The A40I then encapsulates and reorganizes the data before transmitting it to the deck-end computer through the network and fiber optic equipment for display using the host computer software. As part of our study, we will delineate the implementation of the functions required by MCSEM into discrete components, namely the embedded system and the host computer software.

## 3.1 Embedded software development and lens data acquisition

A40I is embedded with the Linux kernel and the software environment mainly includes: Lichee development kit; Qt development kit. The main architecture of the program is shown in Figure 12.

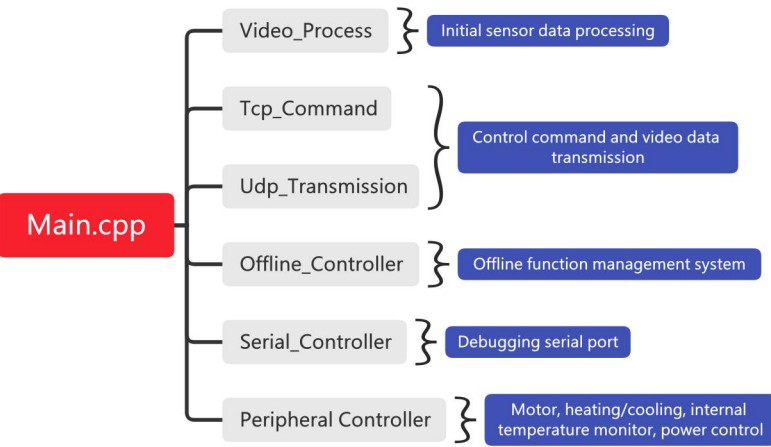

**Figure 12: The main architecture of the program**

In the initial data processing phase, the data provided by the sensor contains a variety of information, including temperature measurement data and other data frames, necessitating the segmentation and repackaging of the received data. This processing stage is carried out in the "Video_Process" thread. To support data transmission efficiently, we utilized a specialized approach that involves employing both TCP and UDP, rather than relying on a single communication channel. TCP is used for establishing a reliable communication channel and facilitating command exchange, while UDP is leveraged for transmitting large volumes of video data at high speeds. The coordination of these communication methods is overseen by two distinct threads: "Tcp_Command" and "Udp_Transmission."

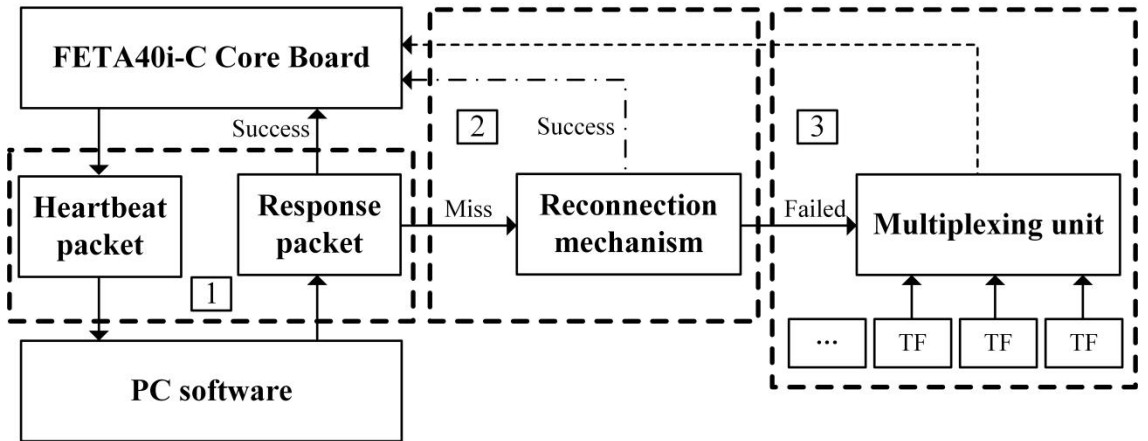

**Figure 13: Disconnection handling mechanism.**

Ocean operations, particularly underwater missions, present significantly greater monitoring challenges compared to land-based operations, inevitably leading to connection disruptions. Therefore, implementing mechanisms to manage such situations and enhance data security becomes essential. The "Offline_Controller" offers a processing approach illustrated in Figure 13. The system employs a heartbeat packet mechanism to verify connection stability, requiring prompt PC responses to each heartbeat signal. When the A40I processor fails to receive a response, it activates the reconnection module to restore PC communication. Persistent disconnections beyond a predefined threshold trigger the system to identify a connection failure and engage the local storage module for data preservation. Once a connection is re-established, the data is automatically re-uploaded. To tackle restricted hardware storage channels during local storage, stacking technology is employed alongside a storage multiplexing

pathway design. The system seamlessly switches to an alternative storage block when the current one is full. This feature allows for the potential deployment of exceedingly large storage units given adequate hardware space. The current configuration employs three 1TB storage units, selected according to operational demands. Notably, this capability sets it apart from similar products like Hikvision's DS-2TD2537T-15/Q. Through this strategic design, the issue of connection interruptions in MCSEM is effectively resolved without requiring manual intervention.

## 3.2 Upper computer software design and temperature calibration

The software is based on the Qt development environment. It integrates the SQLite lightweight database, OpenCV visual library, and plugins such as QCharts and QMultimedia for the development of temperature measurement software. And Figure 14 provides a brief overview of the functions of the PC software.

Since our device only encapsulates the sensor data, the conversion from raw-data to visualized video needs to be done on the PC side, to convert grayscale values to corresponding temperature values, and it is the most important task of the software. According to Planck's law, we can determine the spectral distribution of blackbody radiation:

$$M_\lambda = \frac{c_1}{\lambda^5}(e^{\frac{c_2}{\lambda T}} - 1)^{-1}(w \cdot cm^{-2} \cdot \mu m^{-1}) \tag{3}$$

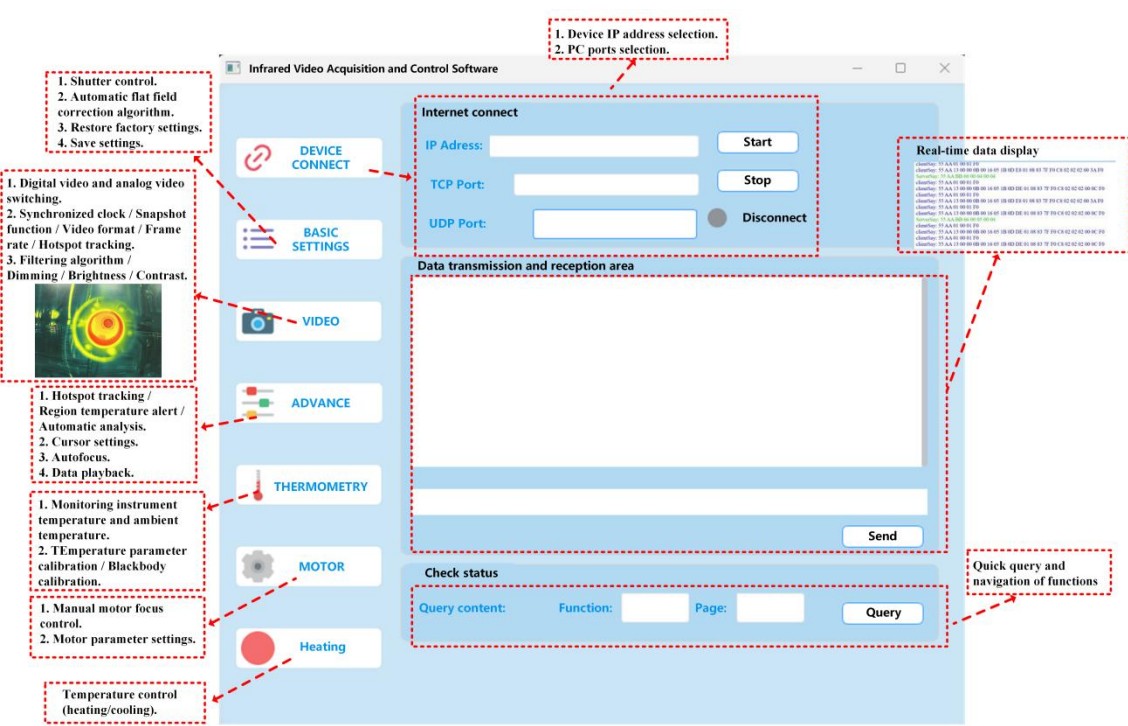

**Figure 14: Introduction to PC Software**

Where $c_1$ is the first radiation constant, and $c_2$ is the second radiation constant. An object with a non-zero absolute temperature T, can emit electromagnetic waves outward, and its radiation intensity satisfies the Stefan-Boltzmann law:

$$M_B = \int_0^\infty M_\lambda \, d\lambda = \sigma T^4 \tag{4}$$

Where $\sigma$ is the Stefan-Boltzmann constant. For an absolute black body with a certain temperature, its spectral radiance has a maximum value, and corresponding to this maximum value, the wavelength of light is according to Wien's displacement law, there is a relationship between the black body temperature T and the peak wavelength , which can be expressed as:

$$\lambda_m \cdot T \approx 2897.8(\mu m \cdot K) \tag{5}$$

Then use $\varepsilon$, commonly known as the emissivity, to characterize the radiation coefficient that varies with material properties and surface conditions:

$$\varepsilon = \frac{M(\lambda, T)}{M_0(\lambda, T)} \tag{6}$$

$M$ means the total radiated power of an object, $M_0$ means the amount of radiation emitted by a black body at the same temperature. Therefore, the radiated energy of a general object satisfies the following equation:

$$M = \varepsilon \sigma T^4 \tag{7}$$

So, as long as the radiated energy of the object is measured and the emissivity of the object is known, the temperature of the
240 object can be calculated as:

$$f(T) = \int_{\lambda_1}^{\lambda_2} \frac{R_\lambda}{\pi} \cdot \frac{c_1}{\lambda^5} \cdot (e^{c_2/\lambda T} - 1)^{-1} d\lambda \tag{8}$$

Where $\lambda_1$ are the lower limit of the spectral response of the detector, and $\lambda_2$ are the upper limit of the spectral response of the detector. At the same time, we need to pay attention to the three aspects of effective radiation received by the thermal detector: target's own radiation, environmental reflected radiation , and atmospheric radiation, so the basic formula for temperature measurement can be represented as follows:

$$f(T_r) = \tau_a[\varepsilon_r f(T_0) + (1 - \varepsilon_r)f(T_u)] + (1 - \tau_\alpha)f(T_\alpha) \tag{9}$$

Where $\varepsilon_r$ the target atmospheric emissivity. When the measured temperature is significantly higher than the ambient temperature, the environmental radiation can be ignored. Similarly, when conducting indoor measurements, atmospheric radiation can be neglected. Therefore, our software has provided API interfaces for these parameters to accommodate different environments and obtain temperature measurements as accurately as possible. In this case, we obtain the relationship between the grayscale average of the thermal images output by the thermal detector and the radiant intensity received by the thermal
detector. By using this relationship, we can measure the temperature by deducing the radiant intensity of the object from the grayscale average of the thermal image displayed by the thermal detector, and then using the relationship between radiant intensity and temperature.

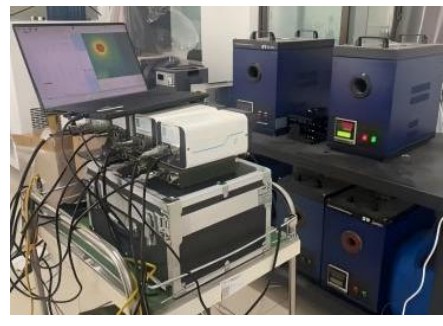 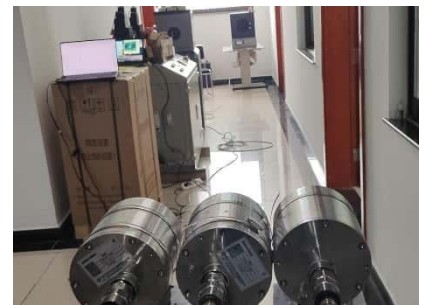

(a) Close-range testing at 1m                    (b) Long-range testing at 5m

**Figure 15: Black body test**

As shown in Figure 15, after configuring the environmental parameters of the software, we conducted tests on the devices using the standard black body. According to the characteristics of the sensor, we divide the temperature measurement range into two
levels, the first level is -20-150℃, and the second level is 0-350℃. At the same time, in order to mitigate the impact of

discrepancies between the displayed temperature and the actual temperature on the data results, we further calibrated the infrared body. The relationship between the actual temperature of the infrared body and the grayscale values is presented in Table 1.

We can clearly observe from the table that there is a certain linear relationship between grayscale values and actual temperature values. Moreover, in the second level, the highest temperature range exceeds 350°C. At 350°C in the second level, the grayscale value is still lower than the grayscale value at 150°C in the first level. However, according to the empirical values in MCSEM, we do not require such a high temperature range. Therefore, we do not focus on grayscale values at higher temperatures.

**Table 1**
The relationship between the actual temperature of the infrared body and the grayscale values at 1m.

| Benchmark temperature | Calibrated temperature | First level | Second level |
|---|---|---|---|
| -20 | -20.5 | 2910.01 | |
| -10 | -10.4 | 3038.12 | |
| 0 | -0.4 | 3257.33 | 3249.85 |
| 10 | 9.6 | 3492.67 | 3303.64 |
| 20 | 19.4 | 3809.17 | 3321.30 |
| 30 | 29.2 | 4125.67 | 3339.53 |
| 40 | 38.9 | 4521.00 | 3397.86 |
| 50 | 48.8 | 4823.23 | 3453.50 |
| 70 | 68.6 | 5705.01 | 3506.80 |
| 100 | 96.9 | 7245.83 | 3634.52 |
| 120 | 116.3 | 8401.92 | 3848.68 |
| 150 | 145.3 | 10500.90 | 4008.41 |
| 200 | 193.9 | | 4300.08 |
| 250 | 242.3 | | 4897.46 |
| 300 | 290.6 | | 5595.70 |
| 350 | 338.9 | | 6403.11 |

Following the aforementioned process, we calibrated multiple instruments. Considering various environmental parameters, we fitted the temperature function and incorporated it into the software's data processing pipeline. After practical testing, we obtained temperature measurement curves at the software level for several devices, as depicted in Figure 16. We have selected three temperature points of particular interest for testing, with the instrument operating at a normal temperature of 50°C (actual 48.8°C), a warning temperature of 150°C (actual 145.3°C), and an upper temperature limit for measurement of 350°C (actual 338.9°C). It can be seen that after calibration, the measurement error of the instrument has fallen within an acceptable range.

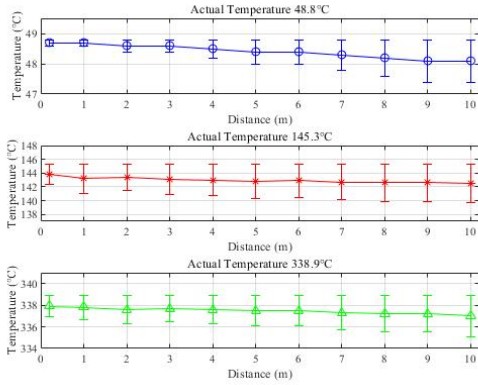

**Figure 16: Temperature vs. distance curve**

## 4 Result

In light of the design highlighted above, our instrument is compared in two dimensions. The first dimension involves a comparison with the original MCSEM monitoring system developed by Wang Meng et al., whereas the second dimension entails a comparison with the well-established explosion-proof infrared temperature measurement system available on the market.

We have innovated the measurement method in our system compared to the traditional MCSEM system by changing from contact point measurement to global surface measurement. This enhancement allows for more intuitive and comprehensive temperature monitoring and early warning capabilities. Additionally, our design upgrades the transmission method to Gigabit Ethernet for voltage, current, and temperature monitoring, in contrast to the traditional serial port transmission, resulting in a significant improvement in transmission speed. Furthermore, we have addressed electromagnetic interference at the circuit level, a feature not present in traditional monitoring systems.

Compared to the commercial HIKVISION company's DS-2TD6537T-25H4LX/W, our system has the ability of secondary correction, which can further calibrate the temperature through parameter settings when used in different environments. Meanwhile, we have a larger data transmission bandwidth than DS-2TD6537T-25H4LX/W, which can meet higher frame rates and larger data packet transmission. In addition to global temperature warning, based on the requirements of MCSEM, we have added region of interest warning, which makes it more convenient to focus on specific locations. Through memory stacking technology, we have achieved at least 3TB of data storage space, greatly extending the offline storage time. In terms of temperature measurement accuracy, we have achieved the same 2% as DS-2TD6537T-25H4LX/W, but it is lower than the contact temperature measurement of 0.5%. However, this is acceptable for MCSEM because we focus on temperature changes in a larger scale range. The detailed technical comparison is shown in Table 2. In conclusion, it can be seen that our design excels in its adaptability to the marine electromagnetic field, enabling better utilization of existing marine electromagnetic equipment (such as optoelectronic composite cables). It can also meet the more demanding environments (offline mechanism, local storage, area of interest alarms, and secondary calibration functionality).

**Table 2**
**Technical Comparison**

| | Traditional MCSEM monitoring system | Non-contact monitoring system | HIKVVISION DS-2TD6537T-25H4LX/W |
|---|---|---|---|
| Measurement temperature | -55 ℃ to +125 ℃ | -20 to 150 ℃ / 0 ~ 350 ℃ | -20 to 550 ℃ |
| Measurement method | single-point contact measurement | **global visual non-contact measurement** | **global visual non-contact measurement** |
| Measurement distance | None | 0.2m~10m | < 200m |
| Secondary correction | none | **yes** | no |
| Data transmission method | serial port | Ethernet | Ethernet |
| Data transmission rate | 115200 bps | **> 1 Gbps** | 10/100 Mbps |
| Alarm function | single-point alarm | **arbitrary interest areas alarm** | fixed area alarm |
| Data reporting interval | 1s | Real-time | Real-time |
| Support for offline storage | yes | yes | yes |
| Offline storage data capacity | 32 GB | **at least 3 TB** | 256 GB |
| Surge protection | none | **yes** | **yes** |
| Temperature measurement accuracy | 0.5% | 2% | 2% |

## 5 Conclusion

This article discusses the security issues of MCSEM and points out the lack of effective security monitoring methods in the development process of MCSEM for high voltage and large current. The article proposes the application of infrared monitoring systems in MCSEM to achieve real-time visualization of structure and temperature. However, the commonly available infrared

devices on the market cannot fully meet the requirements of MCSEM, and there is also a lack of compatible software systems. Hence, starting from the circuit design, we applied electromagnetic interference resistance technology and storage space stacking technology to ensure data stability and security. To adapt to the complex marine environment, we customized variable focal length lenses for infrared sensors and equipped them with high-precision motor systems for control. In terms of embedded

software, a three-stage dropout handling algorithm for maritime communication situations was developed, leveraging TCP and UDP parallel technology for gigabit-level data transmission. Upper computer software was independently developed and data was calibrated based on fundamental theories to ensure reliability. This transformation from point monitoring to area monitoring in MCSEM software has redefined the landscape of monitoring systems in the field from multiple aspects. This paper introduces a new approach for monitoring systems in the field of MCSEM, advocating for the use of non-contact infrared technology over traditional contact-based measurements. These advancements can better cater to offshore operations, ensuring the safety of

equipment and personnel.

## Author contribution

Chentao Wang contributed to the conceptualization, methodology, writing of the original draft, and instrument design; Ming Deng provided the initial idea, supervised the project, and oversaw its administration; Zhibin Ren was responsible for visualization, data curation, and figure preparation; and Meng Wang offered resources, funding acquisition, and overall project

support. All authors have reviewed and approved the final manuscript and agree to be accountable for all aspects of the work.

## Competing interests

The authors declare that they have no conflict of interest.

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
