# Peer review of "Application of Non-Contact Infrared Monitoring Technology in Marine Controlled-source Electromagnetic Transmission System"

_EGUsphere, 2025_

## Author Response (AR1)

**Reviewer 1**

The paper mainly addresses on safety monitoring issue in marine electromagnetic system. Innovative implementation is as follows:

Proposes infrared surface measurement over traditional contact point temperature measurement, not only enlarging parameter monitoring range but also improving flexibility, as demonstrated in temperature calibration to adapt to various configuration in MCSEM system.

Improves system safety performance by applying multi-stage anti-electromagnetic interference design at the circuit level, a feature not present in traditional monitoring systems.

Applies the state-of-the-art technology, such as optical fiber based Gigabit Ethernet transmission, coupled with user friendly interface software, ensuring the system real-time performance as well as safety of equipment and personnel.

What needs to be revised before publication, please reference to the enclosure paper.

**Response to Reviewer 1**

Currently, we have revised the manuscript according to your comments, primarily focusing on the following two aspects:

(1) Carefully revised the grammatical issues you highlighted..

(2) Thoroughly investigated your query regarding Figure 16. Since we manufactured multiple infrared thermometers, each with unique calibration parameters pre-programmed in the software, it's crucial to select the correct device number during operation. The deviation shown in Figure 16 resulted from an incorrect device number selection, which affected not just the second set of measurements. We've consequently repeated the experiments with strict adherence to calibration numbers, reacquired all data, and replotted Figure 16. We sincerely appreciate your meticulous review that helped us perfect the manuscript.

**Reviewer 2**

**1   Discussion on the Applicability of PoE Technology**

Currently, PoE (Power over Ethernet) technology is widely used in network camera devices, but this paper does not mention this solution. Could the authors explain why PoE was not adopted? Was its compatibility with the existing design (e.g., power supply distance, power requirements, or cost factors) evaluated?

**2   Basis for Main Control Chip Selection**

**L85**, the replacement of STM32 with the A40I as the main control chip is mentioned. However, the STM32 series includes multiple performance tiers (e.g., Cortex-M0/M4/M7). Please clarify:

Which specific STM32 model was compared with the A40I?

Were key factors such as real-time performance, peripheral resources, or power consumption considered during the selection process?

**3   File System Optimization Suggestions**

**L90**, when discussing storage capacity, why was a lightweight file system (e.g., FATFS) not considered? This solution is known for significantly lower resource consumption in embedded systems compared to the proposed approach. Please analyze the trade-offs between the two in

terms of read/write speed, power-off protection, and compatibility.

**4 Consistency in Interface Labeling**

**L95**, Figure 4 indicates that the PLUG617 is connected via a Type-C interface, but the diagram does not clearly show this interface. Please verify:

Is this due to simplification in the schematic?

If Type-C is indeed used, the interface circuit design should be supplemented.

**5 Completeness of EMI Design**

In the anti-interference design shown in Figures 6 and 7, the critical role of capacitors in common-mode and differential-mode filtering is not highlighted.

**6 Standardization of Formula Notation**

Some formulas in the text (e.g., $\lambda_1$ and $\lambda_2$) do not correctly use subscript formatting, which may lead to ambiguity. Please ensure consistency in the physical meaning and typesetting of mathematical symbols.

**7 Language Expression Optimization**

**Response to Reviewer 2**

Thank you very much for your detailed feedback. In the early stages of the design, we did consider Power over Ethernet (PoE) as an option. From a technical perspective, PoE technology has evolved over many years and is now at a very mature stage. However, our device was not designed for low power consumption, as its total power requirement reaches several tens of watts, which exceeds the capacity of standard PoE solutions. Typically, a PoE switch supplies power to multiple front-end IP cameras simultaneously. Any failure in the switch's PoE power module would render all connected cameras inoperable, creating an overly concentrated risk. Compared to other power supply methods, PoE technology would increase post-sales maintenance efforts. In terms of safety and stability, a dedicated power supply offers the highest reliability and security.

You are absolutely right to point this out. In our comparison, we were actually referring to the STM32F4 series of MCUs, which were commonly used in previous MCSEM systems. The A40i, on the other hand, is primarily designed for vision-based interactive industrial control products. The A40i meets industrial-grade standards, including compliance with the AEC-Q100 temperature rating. This processor is well-suited for applications requiring 3D graphics and advanced video processing. It offers rich user interfaces, high quality, low power consumption, and a high level of system integration. For our specific application scenario—which involves large-capacity storage and high-volume video data transmission—the A40i is undoubtedly a superior choice over the STM32.

Compared to FATFS (commonly used in embedded systems), Linux offers significant advantages in file management: Advanced File Systems: Supports modern file systems (e.g., ext4, Btrfs) with journaling, crash recovery, and better reliability, while FATFS lacks these features and is prone to corruption. Optimized for large files and high throughput with caching and read-ahead mechanisms, whereas FATFS suffers from fragmentation and slower I/O.

(1) **Functionality:** Includes permissions (user/group/ACL), symbolic/hard links, and network file system support (NFS/Samba), while FATFS has no security controls or advanced features.

(2) **Scalability**: Handles larger files (up to 16TB+) and partitions (1EB+), unlike FAT32's 4GB file limit.

(3) **Development Tools:** Provides robust debugging (e.g., fsck, strace), while FATFS offers minimal troubleshooting support.

FATFS remains suitable for low-resource MCUs (e.g., STM32) or cross-platform compatibility (e.g., USB drives). However, for high-performance, reliability, or complex storage needs, Linux is the superior choice.

4

Thank you very much for your thorough review. This was indeed an oversight on our part. Between PLUG617 and the Baseboard, there is actually a very small adapter board that converts the Type-C interface to an FPC ribbon cable before connecting to the Baseboard. We will update the images in our subsequent submissions.

5

Thank you for pointing out this issue—it helps us further refine the article. In both diagrams, we have included common-mode and differential-mode capacitors to filter out corresponding interference. We will elaborate on this design in our subsequent revisions.

6 and 7

Regarding Questions 6 and 7, we will carefully refine the language and standardize the expression of formulas in our subsequent revisions. Thank you once again for raising these issues.